# A Clinical Prediction Model for Breast Cancer in Women Having Their First Mammogram

**DOI:** 10.3390/healthcare11060856

**Published:** 2023-03-14

**Authors:** Piyanun Wangkulangkul, Suphawat Laohawiriyakamol, Puttisak Puttawibul, Surasak Sangkhathat, Varanatjaa Pradaranon, Thammasin Ingviya

**Affiliations:** 1Department of Surgery, Faculty of Medicine, Prince of Songkla University, Songkhla 90110, Thailand; 2Translational Medicine Research Center, Faculty of Medicine, Prince of Songkla University, Songkhla 90110, Thailand; 3Division of Radiology, Faculty of Medicine, Prince of Songkla University, Songkhla 90110, Thailand; 4Department of Family and Preventive Medicine, Faculty of Medicine, Prince of Songkla University, Songkhla 90110, Thailand; 5Division of Digital Innovation and Data Analytics, Prince of Songkla University, Songkhla 90110, Thailand

**Keywords:** prediction model, breast cancer screening, mammography, breast cancer risks

## Abstract

Background: Digital mammography is the most efficient screening and diagnostic modality for breast cancer (BC). However, the technology is not widely available in rural areas. This study aimed to construct a prediction model for BC in women scheduled for their first mammography at a breast center to prioritize patients on waiting lists. Methods: This retrospective cohort study analyzed breast clinic data from January 2013 to December 2017. Clinical parameters that were significantly associated with a BC diagnosis were used to construct predictive models using stepwise multiple logistic regression. The models’ discriminative capabilities were compared using receiver operating characteristic curves (AUCs). Results: Data from 822 women were selected for analysis using an inverse probability weighting method. Significant risk factors were age, body mass index (BMI), family history of BC, and indicated symptoms (mass and/or nipple discharge). When these factors were used to construct a model, the model performance according to the Akaike criterion was 1387.9, and the AUC was 0.82 (95% confidence interval: 0.76–0.87). Conclusion: In a resource-limited setting, the priority for a first mammogram should be patients with mass and/or nipple discharge, asymptomatic patients who are older or have high BMI, and women with a family history of BC.

## 1. Introduction

According to the Global Cancer Statistics 2020, the incidence of breast cancer has surpassed lung cancer and become the most common type of cancer in females, with a global incidence of newly diagnosed cases of 2.3 million women in 2020, and an estimated 685,000 patients who died of the disease in the same year [1,2]. In Thailand, the age-standardized incidence rate (ASR) of breast cancer was 28.5 cases per 100,000 person-years (PY) in 2012. The ASR of breast cancer in Thailand is expected to increase by 43% from 2012 to 2025 [3]. Changing lifestyles in recent decades, maybe as a consequence of rapid urbanization, explain the increasing modifiable risk factors for cancer. Increasing awareness among at-risk women and improving accessibility to breast cancer screening programs would increase the rate of early diagnosis and allow earlier treatment, thus decreasing mortality [4,5].

Mammography is the current gold standard screening tool for breast cancer as it is the only modality that has been proven to reduce breast cancer-related mortality in various population-based studies [6,7,8]. The American Cancer Society recommends that women between the ages of 45 and 54 years have mammographic screening every year, and every other year thereafter [9] the American College of Radiology (ACR) suggests that all women start annual mammography at age 40; and the U.S. Preventive Services Task Force (USPSTF) and the American College of Physicians (ACP) recommend beginning screening mammographies at age 50 and having them biennially thereafter [10] Each year there are an increasing number of breast cancer screenings, and waiting lists grow longer. To determine the optimal frequency of screening for mammograms by risk factors, mathematical models have been developed. However, these models still have high false positive rates [11,12,13]. Thus, a simple but accurate screening method to identify women at higher-than-average risk would be very beneficial to improve timely breast cancer detection rates.

Gail and Rimer have proposed individualized screening for every woman to assess their lifetime risk. Multiple risk-assessment models, i.e., the RCAPRO, Claus, Breast Cancer Surveillance Consortium (BCSC), and Tyrer–Cuzick models, have been developed which have approximately the same moderate predictive accuracy and good calibration overall [14,15]. To optimize mammography usage, some studies have been conducted that more accurately assess current individual risks or feature improved risk prediction such as a heuristic-based regression model based on age-specific breast cancer risk estimation and annual mammogram screening decision-making [16]. Various breast cancer risk prediction models for Thai women have been created over the past two decades, all of which are used to predict the risk of developing breast cancer within five years. Despite the availability of numerous risk models, however, these have not been extensively implemented to guide routine clinical screenings or predictions based on a first mammogram.

In Thailand, the utilization of mammography examinations is unequal between regions and socioeconomic groups. A survey by the National Statistical Office of Thailand in 2009 showed lower utilization of breast cancer screening in the lower-income group than among women from wealthier families; moreover, healthcare centers and community hospitals in non-municipal or rural areas did not provide mammograms, and people from rural areas needed to be referred to provincial or larger hospitals [17]. In Thailand, screening mammography is opportunity-based and voluntary, which means that the test has not been incorporated into the universal coverage program provided by the National Health Security Office, and an asymptomatic individual who requests a checkup needs to pay for it herself [18]. In addition, the availability of screening machines is limited to larger hospitals where radiologists are also available [19]. For these reasons, it would be useful to have a clinical tool that can accurately assess breast cancer risk in individuals who decide to request breast cancer screening. Although several risk-prediction tools have been developed in recent years, none of these tools use the factors recorded at the patient’s first mammogram, which might lead to selection bias due to knowledge from serial screening or prior mammographic findings [14,20].

In addition, much recent breast cancer prediction research uses genetic information that is frequently unavailable in low- and middle-income countries [21,22,23]. In Thailand, an earlier predictive model was constructed using cross-sectional data from patients registered for mammography in a large center in Bangkok. The model used age, history of contraception, BMI, and menopausal status as independent variables to calculate a risk score. Although the model had high specificity, the sensitivity and discriminating performance were poor and it had limited external validity [24]. Another study, by Chang et al., found more patients positive for cancer among first-visit patients than in follow-up groups in breast cancer screening programs, a finding which supports the importance of prioritizing women registering for a first mammogram, particularly in resource-limited situations [25]. The present study aimed to create a breast cancer prediction tool for individuals registered for a first screening or a diagnostic mammogram study by examining pretest counseling data and final diagnosis data after all sequential diagnostic tests. The resulting model constructed based on these data was intended to be used in prioritizing patients for breast cancer investigations.

## 2. Materials and Methods

### 2.1. Study Design and Setting

This was a retrospective cohort study using data from the Tanyawej Breast Center registry at Songklanagarind Hospital, a tertiary care center and medical school in southern Thailand, from 1 January 2013 to 31 December 2017. Patients who underwent mammography for the first time, whether for screening or diagnosis, were selected to be included in the analysis. Women with a history of invasive breast carcinoma or breast carcinoma in situ, or other types of breast cancer prior to the mammographic study, were excluded (Figure 1). The study protocol was approved by the Office of the Human Research Ethics Committee of the Faculty of Medicine, Prince of Songkla University.

### 2.2. Data Acquisition and Outcome Ascertainment

The data were gathered from structured interviews with each woman performed by trained personnel of the Tanyawej Center. The information collected included demographic data, breast symptoms, history of malignant diseases in the patients and their families, reproductive history, and contraceptive use. The mammographic findings that the center uses are described in the Breast Imaging Reporting and Data System (BI-RADS) protocol [26]. In general, patients with BI-RADS 3 are advised to have a repeat study within one year, while BI-RADS 4 or more indicates an immediate further study or biopsy. To ensure the accuracy of the diagnosis, the clinical follow-up period in all patients is at least five years from their first visit. The medical records and diagnoses based on the International Statistical Classification of Diseases-10 code of each patient were reviewed, and a final diagnosis of breast cancer was based on a pathological report.

### 2.3. Data Management and Statistical Analysis

Data analysis was performed using R Version 3.4.5 (R core team, Vienna, Austria). Baseline characteristics are presented as numbers (%), median (IQR), or mean (± standard deviation), as applicable. For the association analysis, the variables were compared using either Fisher’s exact test, chi-squared test, independent *t*-test, or Wilcoxon’s rank-sum test as appropriate for data type and distribution pattern. To select independent predictive factors associated with a breast cancer diagnosis, inferential statistics were calculated using multiple logistic regression. Sensitivity analysis was undertaken by applying inverse probability weighting (IPW) to select the training dataset, as described by Narduzzi et al. [27], to identify potential selection bias. A *p*-value of < 0.05 was considered statistically significant. To assess the possibility of multicollinearity, the variant inflation factors (VIF) were calculated for each dependent variable, which were generally low, ranging from 1.02 to 2.11, implying a low chance of multicollinearity. Predictive models with 2–5 factors were constructed to be tested with the training dataset which included all patients with complete data. First, the discrimination capabilities of the 2- to 5-factor models to predict breast cancer were quantified using receiver operating characteristic (ROC) curves and areas under the curves (AUCs) using the dataset including all patients. To assess the stabilities of the calculated AUCs and to avoid overfitting, the ROC and AUCs were then calculated using a 10-fold cross-validation method as described in the study by Jung and Hu [28].

## 3. Results

Among the total 61,286 mammograms performed during the study period, 4634 were first mammograms. Breast cancers were diagnosed in 276 (6.0%) patients (Table 1). Of the 4634 women who had had their first study, 822 volunteers (90 breast cancer, 17.7%) were interviewed for further details by a specially trained interviewer. The average age of the participants was 48.9 years. Seventy-two women had a family history of breast cancer. The most common indications for mammography were a palpable breast mass and screening in asymptomatic patients. Age (*p* < 0.001), BMI (*p* < 0.001), and menstruation status (*p* < 0.001) differed significantly between the women with an eventual diagnosis of cancer and those without (Table 2).

### Model Selection

Factors independently associated with a breast cancer diagnosis in the final multiple logistic regression model included age, BMI, family history of breast cancer, and indication for mammography. Age ≥ 50 years had a strong association with breast cancer with an odds ratio (OR) of 5.5 (95% confidence interval (CI): 4.2–7.3). A BMI of ≥ 30 kg/m^2^ had an OR of 2.4 (95% CI: 1.6–3.5). A family history of breast cancer was positively associated with cancer with an OR of 1.5 (95% CI: 1.0–2.2). Presenting with a palpable mass and presenting with nipple discharge demonstrated strong associations with cancer with ORs of 8.9 (95% CI: 6.4–12.6) and 12.9 (95% CI: 6.2–25.2), respectively (Table 3).

When multiple types of prediction models were constructed and evaluated for their performance against a breast cancer diagnosis, the four-factor model, with age, BMI, family history of breast cancer, and indication for mammography, had the highest discriminative performance with an Akaike’s information criterion of 1387.9 and AUCs of 0.82 (95% CI: 0.8–0.9) from the training dataset and 0.78 (95% CI: 0.8–0.8) from the 10-fold cross-validation method (Figure 2, Table 4). If the indications for mammography were omitted from the models, the AUCs ranged between 0.68 and 0.71 in the training dataset and 0.64–0.66 from the 10-fold cross-validation method.

## 4. Discussion

This study aimed to construct a predictive model for prioritizing patients registering for a first mammogram by using multiple logistic regression. Consistent with a previous study analyzing a breast cancer registry in the Thai population [29], older age was a strong risk factor for a finding of breast cancer. Another risk factor for breast cancer in our study was a high BMI. BMI was included in prediction models in earlier studies [24,30], one of which found that a BMI of 30 or more was associated with hormone-positive, pre-menopausal breast cancer [31]. In studies from the Western world, genetic factors have been shown to have a strong influence on breast cancer and have been incorporated into various risk prediction models [32,33]. However, genetic studies are not common in Thailand due to our socioeconomic status. A family history of cancer, especially breast cancer, was then used instead, and our study found that a family history of breast cancer was associated with a significantly higher risk, although at a marginal confidence interval. This finding was consistent with previous studies conducted in Asia [24,29,32,33,34].

Our study also found that patients with a palpable mass and/or abnormal nipple discharge had a significantly higher chance of being diagnosed with breast cancer by a diagnostic mammogram. These findings were in line with previous studies which emphasized that priority in receiving an imaging study should be given to patients with these breast symptoms, especially those with bloody nipple discharge as this was a strong predictor of malignancy [35]. A recent study also found that the presence of a breast mass was associated with a higher chance of finding breast cancer [36]. Our study did not find an association between mastalgia and positive mammography, although this is a common symptom motivating women to ask about a mammogram. A previous study found that mastalgia alone was associated with a very low chance of having cancer and recommended that other clinical factors should be considered together with other symptoms [37].

In this study, our prediction models were created by using the four factors identified in a stepwise logistic regression analysis. When all four parameters, advanced age ≥ 50 years, BMI ≥ 30.0 kg/m^2^, presence of one or more breast symptoms, and family history of cancer, were put into the equation, the AUC at 0.82 (95% CI: 0.8–0.9) was similar to that of the five-parameter equation that included menopausal status. This AUC was higher than the previously reported prediction model in the Thai population that used the four parameters of age, BMI, use of oral contraceptive pills, and menopausal status, of which the final model gave an AUC of 0.65 [24]. A strong point of the aforementioned study was its robustness, as the performance of their equation could be validated in both internal bootstrap and external validation with independent datasets [24].

To simplify prioritizing patients on a mammography waiting list, the two-factor model of our study, with only age ≥ 50 years and the presence of indicating symptoms for mammography, might be preferable in a quick-decision setting as it showed an acceptable performance at an AUC of 0.79 (95% CI: 0.7–0.8). Given the relatively higher sensitivity than the specificity and since only easily-assessed information of age and symptoms are needed, the two-factor model is suitable as a screening tool for large groups, up to the national level of general female populations that may need a mammogram.

Reproductive factors, such as the number of children and age at first childbirth, have been associated with breast cancer in previous studies [38,39]. However, only a relatively small number of Thai women in our study were comfortable with providing such information. Therefore, a practitioner in Thailand cannot rely on having this information. The factors in our model were chosen to avoid a possible history bias in clinical practice while maintaining acceptable accuracy.

Even though the number of mammogram machines is rising, there are still inequities in healthcare utilization. There are many rural areas in Thailand where mammography services are not generally available, and a patient needing or wanting a mammogram needs to travel a long distance to have their breasts screened, and they or other patients often face a long waiting time, which can delay diagnosis and treatment of a potentially serious disease. Using this prediction model, patients with conditions that indicate a need for priority attention will be prioritized for screening. The border provinces of Thailand are an example of this type of situation [19].

The clinical models resulting from this study are simple and avoid the problems of recall bias, invasive tests, or economic prediction, making them practical for implementation on a national scale. However, the empirical results reported herein should be considered in light of some limitations. As the study was conducted in a referral care setting, our patients might reasonably have been expected to have a higher incidence of breast cancer than the general population. Additionally, patients diagnosed with cancer may have a more complete record than those without cancer. Thus, to reduce selection bias, IPW was applied based on the proportion of subjects chosen for the training dataset. Another limitation was that our prediction model was not validated by an external dataset.

## 5. Conclusions

In conclusion, our study constructed a predictive model which can be useful in scheduling mammographies, particularly in resource-limited settings. By using the model, women with a higher chance of having breast cancer can be identified and prioritized when they are registered on a mammography waiting list. This study has the potential to be used in Thailand and other developing countries with healthcare utilization issues. Additionally, recognizing these factors of concern will increase public awareness of breast cancer and promote breast self-examinations, leading to more specific and timely mammogram visits.

## Figures and Tables

**Figure 1 healthcare-11-00856-f001:**
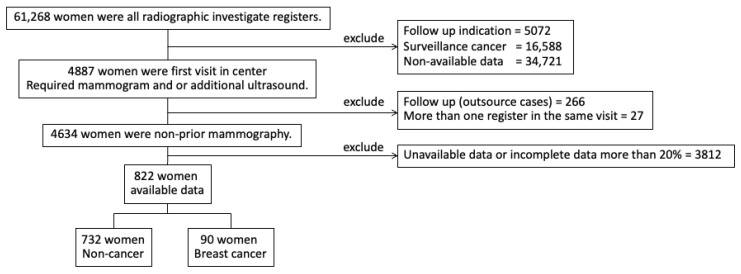
Participant enrolment.

**Figure 2 healthcare-11-00856-f002:**
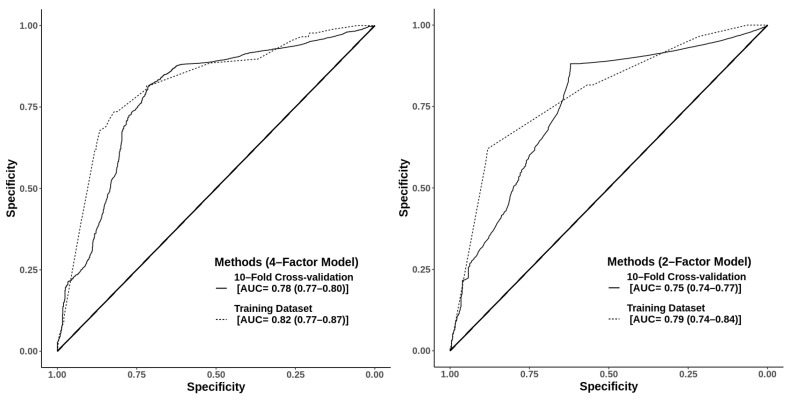
The 4-factor model using age ≥ 50 years, body mass index ≥ 30 kg/m^2^, history of breast cancer in the family, and symptomatic indication for mammography with an Akaike information criterion (AIC) of 1387.9 and areas under the receiver operating characteristic curve of 0.82 (95% confidence interval (CI): 0.77–0.87) and 0.78 (95% CI: 0.77–0.80) from the training dataset and 10-fold cross-validation, respectively. The 2-factor model using age ≥ 50 years and symptomatic indication for mammography, with an AIC of 1404 and areas under the receiver operating characteristic curve of 0.79 (95% CI: 0.74–0.84) and 0.75 (95% CI: 0.74–0.77) from the training dataset and 10-fold cross-validation, respectively.

**Table 1 healthcare-11-00856-t001:** Indication and final diagnosis for women at their first mammogram visit.

Indication	All Datasets				Analyzed Datasets			
	Total	Non-Cancer	Cancer	*p*-Value	Total	Non-Cancer	Cancer	*p*-Value
	N = 4634 (%)	N = 4358 (%)	N = 276 (%)		N = 822 (%)	N = 732 (%)	N = 90 (%)	
Screening	1840 (39.7)	1816 (41.7)	24 (8.7)	<0.001 ^γ^	330 (40.1)	316 (43.2)	14 (15.6)	<0.001 ^γ^
Symptom								
Mass	2063 (44.5)	1831 (42)	232 (84.1)		383 (46.6)	313 (42.8)	70 (77.8)	
Nipple discharge	77 (1.7)	68 (1.6)	9 (3.3)		22 (2.7)	18 (2.5)	4 (4.4)	
Pain	576 (12.4)	570 (13.1)	6 (2.2)		76 (9.2)	74 (10.1)	2 (2.2)	
Other	78 (1.7)	73 (1.7)	5 (1.8)		11 (1.3)	11 (1.5)	0 (0)	

Diagnosis was confirmed using a biopsy only in suspected cancer patients. ^γ^ Comparison between non-cancer and cancer groups, based on the chi-squared test.

**Table 2 healthcare-11-00856-t002:** Baseline characteristics and demographic data of the analyzed dataset according to breast cancer diagnosis.

Characteristic	Total N = 822	Non-Cancer N = 732	Cancer N = 90	*p*-Value
Age, years	49 (41, 58)	48 (40, 57)	55.5 (50, 61.8)	<0.001 ^ϯ^
BMI, kg/m^2^	23.1 (20.4, 25.8)	22.9 (20.3, 25.6)	24.4 (21.5, 27.3)	0.001 ^ϯ^
Religion; *n* (%)				0.782 ^γ^
Non-Muslim	744 (91.9)	662 (91.7)	82 (93.2)	
Muslim	66 (8.1)	60 (8.3)	6 (6.8)	
History of other types of cancer; *n* (%)				0.319 ^£^
No	776 (94.5)	694 (94.8)	82 (92.1)	
Yes	45 (5.5)	38 (5.2)	7 (7.9)	
Family history of breast cancer; *n* (%)				0.156 ^γ^
No	748 (91.2)	670 (91.8)	78 (86.7)	
Yes	72 (8.8)	60 (8.2)	12 (13.3)	
Family history of ovarian cancer, *n* (%)				1 ^£^
No	801 (97.6)	713 (97.5)	88 (97.8)	
Yes	20 (2.4)	18 (2.5)	2 (2.2)	
**Reproductive history**				
Menarche; age	13 (12, 15)	13 (12, 15)	14 (12, 15)	0.743 ^ϯ^
Menstrual status; *n* (%)				<0.001 ^γ^
Pre-menopause; (%)	451 (55.2)	417 (57.4)	34 (37.8)	
Post-menopause; (%)	366 (44.8)	310 (42.6)	56 (62.2)	
Age at menopause; year	48 ± 5.9	49 (45, 52)	50 (48, 52)	0.124 ^ϯ^
Oophorectomy; *n* (%)				0.428 ^ϯ^
No surgery	733 (89.3)	657 (89.8)	76 (85.4)	
Unilateral	24 (2.9)	20 (2.7)	4 (4.5)	
Bilateral	64 (7.8)	55 (7.5)	9 (10.1)	
**External Hormone use**				
History of contraception use; *n* (%)				0.287 ^ϯ^
Never used	490 (65.3)	440 (66)	50 (60.2)	
OCP used	219 (29.2)	189 (28.3)	30 (36.1)	
No OCP used	41 (5.5)	38 (5.7)	3 (3.6)	
Hormonal therapy; *n* (%)				0.071 ^γ^
Never used	699 (85.1)	617 (84.3)	82 (92.1)	
Ever used	122 (14.9)	115 (15.7)	7 (7.9)	
Indication; *n* (%)				<0.001 ^γ^
Screening/asymptomatic	330 (40.1)	316 (43.2)	14 (15.6)	
Mass	383 (46.6)	313 (42.8)	70 (77.8)	
Nipple discharge	22 (2.7)	18 (2.5)	4 (4.4)	
Pain	76 (9.2)	74 (10.1)	2 (2.2)	
Other	11 (1.3)	11 (1.5)	0 (0)	

^ϯ^ Rank-sum test, ^γ^ chi-squared test, ^£^ Fisher’s exact test; BMI: body mass index; OCP: oral contraception. All continuous data are presented as median (IQR) unless stated otherwise.

**Table 3 healthcare-11-00856-t003:** Univariable and multivariable logistic regression for breast cancer diagnosis.

Variable	Univariate Analysis			Multivariate Analysis	
	Crude Odds Ratio (95% CI)	*p*-Value	Coeff	Adjusted Odds Ratio (95% CI)	*p*-Value
Age, years					
<50	1		0	1	
≥50	3.2 (2.5–4.2)	<0.001	1.7	5.5 (4.2–7.3)	<0.001
BMI, kg/m^2^					
≤23	1		0	1	
23–29	1.4 (1.1–1.8)	0.009	0.2	1.2 (0.9–1.6)	0.1406
≥30	3.2 (2.2–4.6)	<0.001	0.9	2.4 (1.6–3.5)	<0.001
Family history of breast cancer					
No	1		0	1	
Yes	1.4 (1.0–2.0)	0.055	0.4	1.5 (1.0–2.2)	0.0439
Breast symptom					
Screening/asymptomatic	1		0		
Mass	5.5 (4.1–7.7)	<0.001	2.2	8.9 (6.4–12.6)	<0.001
Nipple discharge	7.8 (3.9–14.8)	<0.001	2.6	12.9 (6.2 -25.2)	<0.001
Pain	0.5 (0.2–1.0)	0.063	−13.0	0.5 (0.2–1.1)	0.1141
Others	0.0 (NA)	0.966	−0.6	0.0	0.9639

95% CI: 95% confidence interval.

**Table 4 healthcare-11-00856-t004:** Comparing Akaike information criterion and area under the receiver operating characteristic curve values of models with varying factors.

Factor Model	Variables	AIC	AUC (95% CI)	
			Training	10-Fold Cross Validation
5 factors	age, BMI, menstrual status, family history of breast cancer, indication	1380.2	0.82 (0.8–0.9)	0.78 (0.8–0.8)
4 factors	age, BMI, family history of breast cancer, indication	1387.9	0.82 (0.8–0.9)	0.78 (0.8–0.8)
3 factors	age, BMI, indication	1388.7	0.81 (0.7–0.9)	0.77 (0.8–0.8)
2 factors	age, indication	1404.0	0.79 (0.7–0.8)	0.75 (0.7–0.8)
**Factors without indication**				
4 factors	age, BMI, family history of breast cancer, menstrual status	1637.6	0.71 (0.6–0.8)	0.66 (0.6–0.7)
3 factors	age, BMI, family history of breast cancer	1643	0.68 (0.6–0.8)	0.65 (0.6–0.7)
2 factors	age, BMI	1644.2	0.68 (0.7–0.8)	0.64 (0.6–0.7)

AIC: Akaike information criterion; AUC: area under the receiver operating characteristic curve; BMI: body mass index.

## Data Availability

Not applicable.

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
