# Peer review of "A Clinical Prediction Model for Breast Cancer in Women Having Their First Mammogram"

_healthcare, 2023, doi:10.3390/healthcare11060856_

Round 1
Reviewer 1 Report
As a retrospective cohort study, this manuscript aimed to construct a prediction model for BC, based on Clinical parameters. Significant risk factors be found as age, BMI, family history of BC, and indicated symptoms.
In the manuscript, some issues are not clear, and the manuscript needs to be revised.
1. The AUC was calculated, but in the manuscipt, there is no description about the training set and validation dataset.
2. For a research paper, some of the principles of the model need to be introduced in the manuscript.
3. When performing model validation, how to eliminate bias in data selection
4. It is recommended to add external validation
This manuscript aimed to construct a prediction model for BC in women scheduled for their first mammography. The model is built based on some Clinical parameters and the indication of mammography. As I know, for breast cancer, many works have been published about the radiomics research of the mammography data. These papers included the prediction of the risk, or risk assessment (Adaptive decision-making of breast cancer mammography screening: A heuristic-based regression model; Breast Cancer Risk Assessment: Calculating Lifetime Risk Using the Tyrer-Cuzick Model; Improving breast cancer risk prediction by using demographic risk factors, abnormality features on mammograms and genetic variants, etc.) This work builds the model based on some readily available indicators, and it is beneficial in situations where resources are limited. And the conclusions of this work are consistent with the evidence and arguments presented in their manuscript. But, I think the author should pay more attention to the information from the medical images. The results of many published works have proven that the accuracy besed on images is satisfactory, especially for the deep learning model. The description of their methods is somewhat simplified. Some issues need to be considered. 1. for the model proposed in this manuscript, the AUC curve is based on the training set? or validation set? There is no information about the dataset division. 2. for a prediction model, how the author eliminate bias in data selection, CV? or any other methods? A detailed introduction to the dataset is needed in the manuscript. 3. if no CV is used, maybe the author should use external validation to show the performance of the model.
Reviewer 2 Report
Summary: The current study argues for a risk prediction model to stratify subjects for a first time mammography. The authors use a logistic regression model to evaluate the impact of correlates on the risk of testing positive for breast cancer. They summarize their results by identifying that BMI, older age and nipple discharge correlate highly with breast cancer.
Comments:
Overall, the paper has no major flaws. I have some comments/suggestions that need to be addressed.
1. Please include the multi-variate odds ratios in a table to see the effects of the covariates.
2. Have the authors looked at the correlations between the factors using a method like VIF. It would be interesting to see if there is co-linearity in the models.
3. I see from the study design that a lot of subjects were excluded from the study due to incomplete data. Since the authors built 2 factor and 3 factor models, could some of these incomplete subjects used for validation purposes?
4. Would it be possible for the authors to try interaction effects in the logistic regression model?
5. The conclusion section needs to be expanded to talk about how these type of models can be potentially applied in the field. Examples could be:
-Are there regions in the country where the authors think this sort of modeling approach would be helpful?
-Given the true-positive and false-positive rate of the model, what population size would the authors propose for an effective use of this model in the field?
-How can data collection policies improve the use of similar proposed models? For example, Age and BMI information might be available at population level nationally. Is this enough to apply this model in the field at a national level?
Reviewer 3 Report
The authors proposed a prediction model for BC in women scheduled for their first mammography. The model is well-implemented, and the manuscript is well-presented. The model is based can be available for so many low-income communities. However, I have minor suggestions:
-The sensitivity compared to the specificity is relatively lower. and both measures and low, can you discuss the low performance for these measures?
-The conclusion can have more statements. 1-2 statements about the advantages and limitations ( e.g. economic prediction model and no invasive approaches, and low-performance measurements that can be enhanced) could be added.
-For the following statement, you need to refer to recent work with BC prediction based on genetic information. I suggest highlighting/referring to PMID: 35205681 and/or PMID: 30972106.
-Line 56: "Also, most prediction 56 tools include genetic information, which is not generally available in the context of low to 57 middle-income countries."
Round 2
Reviewer 1 Report
I think the current manuscript is suitable for receiving and publishing